# Cross sectional study of Twitter (X) use among academic anesthesiology departments in the United States

**Michael Mazzeffi**[1]*, **Lindsay Strickland**[2], **Zachary Coffman**[1‡], **Braden Miller**[2‡], **Ebony Hilton**[1‡], **Lynn Kohan**[1‡], **Ryan Keneally**[3‡], **Peggy McNaull**[1‡], **Nabil Elkassabany**[1‡]

1 Department of Anesthesiology, University of Virginia School of Medicine, Charlottesville, Virginia, United States of America, 2 Public Health Sciences, University of Virginia School of Medicine, Charlottesville, Virginia, United States of America, 3 Department of Anesthesiology and Critical Care Medicine, George Washington University School of Medicine and Health Sciences, Washington, District of Columbia, United States of America

☯ These authors contributed equally to this work.
‡ ZC, BM, EH, LK, RK, PM and NE also contributed equally to this work.
* mmazzeff@uvahealth.org

**Data Availability Statement:** The dataset used for the analysis is publicly available at OSF: DOI 10.17605/OSF.IO/B7H5G.

## Abstract

Twitter (recently renamed X) is used by academic anesthesiology departments as a social media platform for various purposes. We hypothesized that Twitter (X) use would be prevalent among academic anesthesiology departments and that the number of tweets would vary by region, physician faculty size, and National Institutes of Health (NIH) research funding rank. We performed a descriptive study of Twitter (X) use by academic anesthesiology departments (i.e. those with a residency program) in 2022. Original tweets were collected using a Twitter (X) analytics tool. Summary statistics were reported for tweet number and content. The median number of tweets was compared after stratifying by region, physician faculty size, and NIH funding rank. Among 166 academic anesthesiology departments, there were 73 (44.0%) that had a Twitter (X) account in 2022. There were 3,578 original tweets during the study period and the median number of tweets per department was 21 (25th-75th = 0, 75) with most tweets (55.8%) announcing general departmental news and a smaller number highlighting social events (12.5%), research (11.1%), recruiting (7.1%), DEI activities (5.2%), and trainee experiences (4.1%). There was no significant difference in the median number of tweets by region (P = 0.81). The median number of tweets differed significantly by physician faculty size (P<0.001) with larger departments tweeting more and also by NIH funding rank (P = 0.005) with highly funded departments tweeting more. In 2022, we found that less than half of academic anesthesiology departments had a Twitter (X) account, and the median number of annual tweets per account was relatively low. Overall, Twitter (X) use was less common than anticipated among academic anesthesiology departments and most tweets focused on promotion of departmental activities or individual faculty. There may be opportunities for more widespread and effective use of Twitter (X) by academic anesthesiology departments including education about anesthesiology as a specialty.

**Funding:** The author(s) received no specific funding for this work.

**Competing interests:** The authors have declared that no competing interests exist.

## Introduction

Social media platforms including Facebook, LinkedIn, and Twitter (recently renamed X), are used by anesthesiologists for networking, dissemination of research, faculty promotion, and medical education [1, 2]. Social media was also used by many anesthesiology departments for residency recruitment and virtual open houses during the coronavirus disease-2019 (COVID-19) pandemic [3]. Potential benefits of social media use include greater access to medical students and job applicants, online remote collaboration, promotion of faculty and their academic outputs, and professional growth opportunities for trainees and faculty [4]. Social media also offers an opportunity for professional education and broader education of the public about medical specialties.

Twitter (now X) is one of the most influential social media outlets in the United States. Its platform allows for 280 (originally 140) character messages to be "tweeted". Currently there are over 350 million users, although the amount of activity per user is variable. Images, videos, and web links can be added to tweets and "hashtags" are used to group topics of interest. In a recent study of over 1 trillion tweets from 2016 to 2020, there were 6.4 million tweets from 39,000 physicians [5]. This represented less than 0.001% of all tweets; however, the number of physicians tweeting has doubled in five years [5]. In a 2021 survey of academic surgical departments, 25% had a Twitter (X) account with the median number of tweets being 314 and median account age being 3.5 years [6].

To our knowledge, there is a paucity of research about Twitter (X) use in academic anesthesiology departments. Better understanding current Twitter (X) use could help academic anesthesiology departments to develop better communication and education strategies for the future. Also, understanding tweet content may highlight common challenges and opportunities facing academic anesthesiology departments. The primary aim of our study was to characterize contemporary Twitter (X) use by academic anesthesiology departments in the United States. Secondarily, we sought to understand departmental characteristics that were associated with the number and type of tweets (e.g. region, physician faculty size, and research funding rank). Our primary hypothesis was that Twitter (X) would be used by most academic anesthesiology departments in the United States and that the number of tweets would vary by region, physician faculty size, and National Institutes of Health (NIH) research funding rank.

## Methods

### Study design

The University of Virginia Institutional Review Board determined that the study was not human subjects research and waived the requirement for written informed consent. The study design was a cross-sectional study of academic anesthesiology departments' Twitter (X) use in 2022. Academic anesthesiology departments were defined as those that had an approved core anesthesiology residency program in 2022. The list of approved academic anesthesiology programs was obtained from the Accreditation Council for Graduate Medical Education (ACGME) webpage [7].

Twitter (X) accounts for individual anesthesiology departments were identified on departmental webpages and by free text Google search where the department's name was searched along with the words "Twitter account". Only departmental Twitter (X) accounts were included in the analysis. Residency program accounts, division accounts, and individual faculty member accounts were excluded and were beyond the scope of the analysis.

### Tweet review and classification

Tweets from individual departments were collected using Vicinitas Twitter (X) analytics (https://www.vicinitas.io/). Original tweets were included, while retweets and quote tweets

were excluded from the analysis. Classification of tweets into pre-specified content categories was manually performed by two study authors who read individual tweets and categorized them. The classification groups for tweets were 1) research and publications, 2) diversity, equity, and inclusion (DEI), 3) training program related, 4) recruiting, 5) social event related, 6) general departmental news, and 7) other. The analytic dataset that was created for the study is available to readers as a S1 File and is compliant with all institutional requirements for data security.

### Statistical analysis

Statistical analysis was performed using SPSS version 28.0 (SPSS Inc., Armonk, NY USA). Summary statistics were calculated and reported as the median and interquartile range or number and percent. Box and whisker plots were created to show number of tweets after stratification by department region, faculty size, and 2022 research funding rank from the Blue Ridge Institute for Medical Research (Horse Shoe, NC USA). Region classification was based on consolidated United States census regions. Faculty size was classified as small (1–40), medium (41–80), large (81–120), and very large (121 or greater) and was determined using departmental webpage data. Research ranks from the Blue Ridge Institute for Medical Research were classified as top 25, 26–54, or not ranked. The Kruskal Wallis Test was used to test for significant differences between groups. A word cloud was made using software from Vizzlo (Leipzig, Germany) (https://vizzlo.com) to visually display the departments that had the largest number of tweets during the year.

As a secondary analysis, we analyzed tweet content after stratifying by department characteristics. First, box and whisker plots were made displaying the number of research and publication tweets after stratifying by NIH research funding rank. Second, box and whisker plots were made displaying the number of DEI tweets after stratifying by region and faculty size. The Kruskal Wallis Test was used to test for significant differences between groups. P values <0.05 were considered statistically significant for all tests. Inter-observer reliability for classification of tweet content was estimated using Cohen's Kappa statistic, where 30 tweets were independently classified by two observers and the Kappa statistic was calculated. The Strengthening the Reporting of Observational Studies in Epidemiology (STROBE) checklist was referenced in preparing the manuscript.

### Results

Among 166 academic anesthesiology departments, there were 73 (44.0%) that had a Twitter (X) account in 2022. There were 3,578 original tweets (Table 1) during the year and the median number of tweets per department was 21 (25th-75th = 0, 75). The majority (55.8%) of tweets announced general departmental news, while the next most common types of tweets were social event related (12.5%) and research/publication related (11.1%). A small percentage of tweets were related to DEI (5.2%) and trainee experiences (4.1%). Cohen's Kappa statistic was calculated to be 0.87 for classification of tweet content. Forty-five percent of programs had 10 or fewer tweets during the year, while 13.7% of programs had more than 100 tweets. Fig 1 shows the 30 departments with the most tweets. Departments that tweeted the most are displayed in the largest font and are most central.

Table 2 lists characteristics of the 73 departments that had active Twitter (X) accounts in 2022. The largest number of departments were from the Southern United States (38.4%), while the smallest number (13.7%) were from the Western United States. The majority of departments (52.1%) had more than 80 faculty members with a small number of departments

**Table 1. Tweets from academic anesthesiology programs in 2022.**

| Variable | Median (Q1, Q3) or n (%) |
|---|---|
| Number of tweets per active account | 21 (0, 75) |
| Number of tweets per active account stratified | |
| 0–10 | 33 (45.2) |
| 11–50 | 16 (21.9) |
| 51–100 | 14 (19.2) |
| 101 or more | 10 (13.7) |
| Number of tweets per active account by content | |
| Research/publications | 0 (0, 6) |
| Diversity, equity, inclusion | 0 (0, 4) |
| Training program related | 0 (0, 2) |
| Recruiting | 1 (0, 4) |
| Social event related | 2 (0, 12) |
| General program news | 10 (0, 37) |
| Other | 0 (0, 2) |
| Tweet content stratified | |
| Research/publications | 397 (11.1) |
| Diversity, equity, inclusion | 186 (5.2) |
| Trainee experience | 146 (4.1) |
| Recruiting | 254 (7.1) |
| Departmental social event | 448 (12.5) |
| Program news (match, speaker, thank you) | 1998 (55.8l) |
| Other | 149 (4.2) |

N = 73 programs and 3,578 total tweets

(17.8%) having 40 or fewer faculty. The majority of departments (54.8%) were ranked in the top 54 NIH funded departments in the country.

Fig 2 shows the total number of tweets after stratification by region (A), faculty size (B), and NIH funding rank (C). There was no significant difference in the median number of tweets by region (P = 0.81). The median number of tweets differed significantly by faculty size

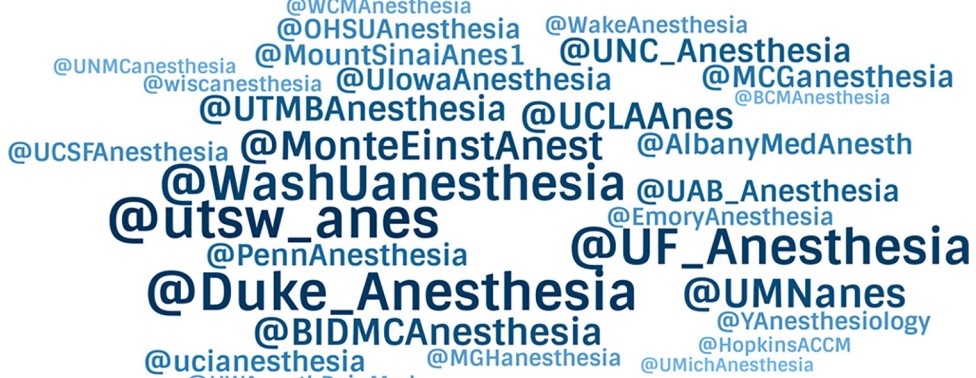

**Fig 1. Figure shows academic anesthesiology department handles with the most Twitter (X) activity.**

**Table 2. Characteristics of programs with active Twitter (X) accounts.**

| Variable | n (%) |
|---|---|
| Region | |
| West | 10 (13.7) |
| Midwest | 15 (20.5) |
| South | 28 (38.4) |
| North East | 20 (27.4) |
| Faculty size | |
| 1–40 | 13 (17.8) |
| 41–80 | 22 (30.1) |
| 81–120 | 21 (28.8) |
| 121 or more | 17 (23.3) |
| Residency size | |
| 1–30 | 11 (15.1) |
| 31–60 | 32 (43.8) |
| 61–90 | 20 (27.4) |
| 91 or more | 10 (13.7) |
| 2020 NIH funding rank (Blueridge Institute) | |
| Ranked in top 25 | 21 (28.8) |
| Ranked 26–54 | 19 (26.0) |
| Not ranked | 33 (45.2) |

N = 73 programs

NIH = national institute of health

(P<0.001) with larger departments tweeting more and also by NIH funding rank (P = 0.005) with highly funded departments tweeting more.

Fig 3 shows the number of research/publication tweets after stratification by NIH funding rank (A). It also shows the number of DEI tweets after stratification by region (B) and faculty size (C). The number of research/publication tweets differed significantly by NIH funding rank (P = 0.006) with more highly funded departments having more research/publication related tweets. The number of DEI tweets did not differ significantly by region (P = 0.96), but did differ significantly by faculty size (P<0.001) with larger departments having more DEI tweets.

## Discussion

In a descriptive study of Twitter (X) use by academic anesthesiology departments in the United States, we found that less than half of departments (44%) had a Twitter (X) account in 2022. Larger departments and departments with more NIH funding used Twitter (X) the most. The total number of original tweets in 2022 was 3,578 and almost half of departments with a Twitter (X) account had 10 or fewer tweets, while 10 departments tweeted more than 100 times. Use by academic anesthesiology departments was slightly less than use by general Twitter (X) users, who tweet approximately 2 times per month [8]. The majority of tweets (55.8%) were related to general departmental news, while the next most common themes were social events (12.5%), research/publications (11.1%), and recruiting (7.1%). Twitter (X) was not used much by academic anesthesiology departments for teaching or to disseminate information about the specialty of anesthesiology to the public.

Social media use has increased rapidly in medicine, particularly during the COVID-19 pandemic. In a recent systematic review, the authors found an increasing number of medical

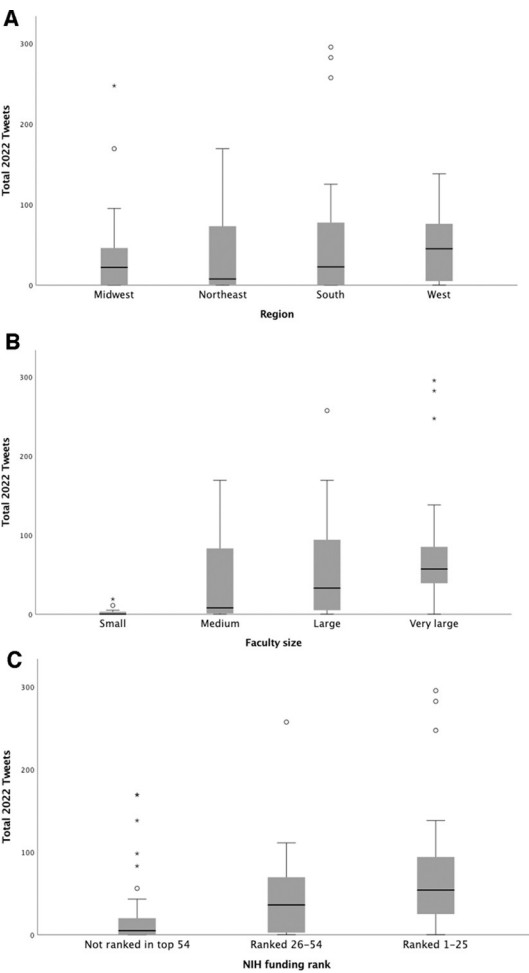

**Fig 2.** Box and whisker plots of number of total tweets after stratification by A) region, B) faculty size, and C) NIH funding rank.

publications related to social media use from 2006 to 2020 [9]. Common themes in social media use include dissemination of health information, combat of misinformation, mobilization of social resources, facilitation of health related research, professional development, exchange of emotional support, and development of healthcare networks [9]. Twitter (X) is one of the most highly used social media platforms by healthcare professionals in the United States, and is popular because it allows interfacing between the healthcare and non-healthcare community [10]. Twitter (X) creates a space for physicians, researchers, and patients to access the most up-to-date research and health information, as well as exchange ideas [10]. Twitter also increases the number of persons that can participate in medical discourse, allowing multiple medical professionals to share their expertise on a "level playing field", and allowing for rapid dissemination of new biomedical research findings [10]. Twitter (X) has been used in dentistry/oral surgery to promote oral health behaviors, to disseminate evidence based practices via chats and journal clubs, and to disseminate information about cosmetic dentistry [11, 12]. Many nurses used Twitter (X) extensively during the COVID-19 pandemic to highlight their difficult working conditions, to advocate for needed resources, and to convey personal and emotional experiences from their work [13].

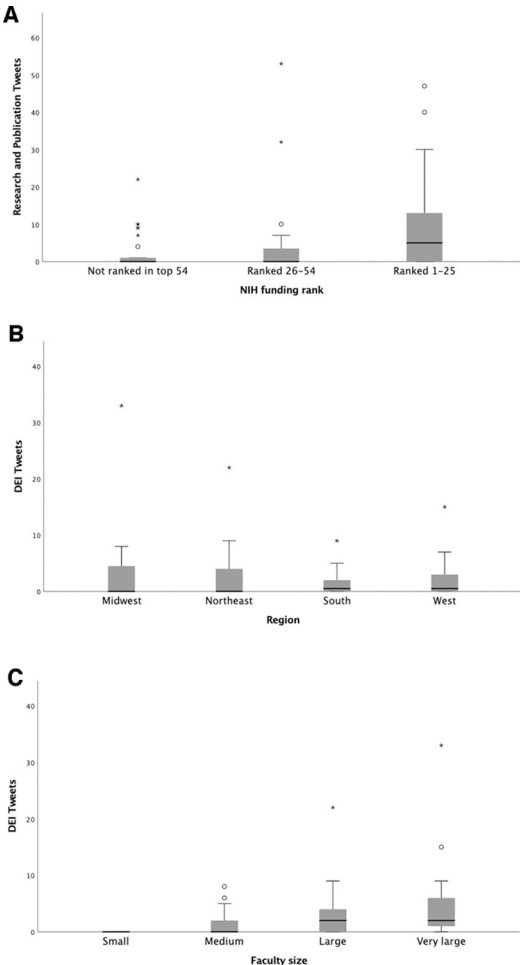

**Fig 3. Figure showing box and whisker plots of number of research/publication tweets after stratification by A) NIH funding and number of DEI tweets after stratification by B) region and C) faculty size.**

Multiple social media platforms, including Twitter (X) are used for medical education and are associated with improved knowledge, learner attitudes, and skills [14]. Studies suggest that social media can improve learner engagement, facilitate learner feedback, and enhance collaboration [14]. Social media has the potential to enhance the public's knowledge about various medical specialties and what they do. Most social media platforms are widely available to the public, eliminating traditional barriers to medical information. Notably, social media postings are not peer reviewed or checked for accuracy, and hence there are risks for misinformation that need to be considered.

There are a few previously published studies of anesthesiologists' and anesthesiology departments' use of social media, and specifically Twitter (X). One recent study analyzed the use of Twitter by anesthesiologists at national anesthesiology conferences [15]. In this study, there were 63,180 conference related tweets and physicians were top influencers at 8 of the 14 conferences [15]. In a second study that compared Twitter (X) use between the American Society of Anesthesiologists (ASA) and American Association of Nurse Anesthetists, the authors found that the ASA had a more active Twitter (X) presence [16]. The regional anesthesiology community has been highly active in its Twitter (X) use, and has produced a large number of

tweets related to education about peripheral nerve blocks, opioid sparing analgesia, and other postoperative pain management strategies [17].

Twitter (X) use appears to be relatively high among other groups of physicians, including surgeons and emergency medicine physicians who have disseminated knowledge from academic conferences and have used Twitter (X) as a platform for communication/collaboration during the COVID-19 pandemic [18, 19]. An analysis of tweets from surgeons found that there was disproportionate use among a small number of users who had many followers, while the majority of accounts had low levels of activity [20]. These findings are consistent with what we observed in our own study, where a few academic anesthesiology departments used Twitter (X) frequently, while most departments had a lower level of use.

Our study demonstrates that a fairly large number of academic anesthesiology departments (N = 73) had a Twitter (X) account in 2022, but overall Twitter (X) use was relatively low with approximately half of departments having less than 1 tweet per month on average. Departments that tweeted more tended to be large and have more NIH funding, which implies availability of more resources and potentially more attention to non-clinical aspects of the academic mission. Larger departments are likely to have more administrative support and in some cases may have a dedicated administrator for communications and or public relations. Factors that may limit Twitter (X) use in anesthesiology departments include lack of administrative personnel, fears related to negative tweet interpretation, institutional regulations, and a lack of expertise in Twitter (X) use. At least one study suggests that non-physicians have more professionalism and confidentiality concerns about health-related Twitter (X) use than physicians [21]. These concerns are in part validated by a study of over 12,000 anesthesiologists' Twitter (X) accounts where 10% of accounts had at least one tweet with a breach of confidential patient information [22]. There are also concerns about harassment from fellow Twitter (X) users with differing viewpoints, and this may lead some departments to avoid Twitter (X) altogether [23].

Our study helps identify opportunities for future Twitter (X) use, and more broadly for social media use in our specialty. For example, we found few tweets that described departmental values, trainee experiences, DEI initiatives, or education about anesthesiology and its related disciplines (e.g. pain medicine, critical care). Future tweets related to these topics may help academic anesthesiology departments with branding and communication of their values and culture to prospective trainees and employees. Additionally, there were few tweets describing the services that physician anesthesiologists provide for patients during their surgical experience. The latter is a major opportunity, in our opinion, given that many patients do not understand the role of anesthesiologists in their care [24]. Most patients have very brief interactions with their anesthesiologist and many do not meet their anesthesiologist or fully understand their role until the day of surgery. Twitter (X) could be used to describe anesthesiologists' role in reducing anxiety, limiting pain, enhancing recovery, and providing critical care after surgery. Twitter (X) was used to promote research/publications to only a small degree with only 11% of tweets being related to this topic. However, departments with more NIH funding tweeted more frequently about their research/publications. Tweets about publications are used to calculate some research impact scores, including the Altmetric attention score (www.almetric.com), which may incentivize departments to tweet about their research/publications [25, 26].

Our study has a number of limitations. First, we only analyzed tweets from academic anesthesiology departments. Hence, our results cannot be generalized to all anesthesiology practices including large private practices. Second, our study did not analyze the Twitter (X) activity of anesthesiology residency and fellowship programs, as well as individual anesthesiologists. Third, we did not survey departments to understand their motivations for using Twitter

(X) or concerns about using it. Fourth, we did not analyze demographics of faculty, which may have impacted departmental tweet content. Fifth, we did not analyze other social media platforms that are used by anesthesiologists such as Facebook, LinkedIn, Medscape, Free Open Access Medical Education (FOAM), or Doximity. Finally, we only analyzed data from one year, which occurred near the end of the COVID-19 pandemic. This may represent a skewed reflection of overall Twitter (X) use by academic anesthesiology departments during the last several years. With Twitter's rebranding as X in July of 2023, it is possible that the number of anesthesiology programs using the platform will change significantly over time.

## Conclusions

In summary, in a descriptive study of academic anesthesiology departments' Twitter (X) use in 2022, we found that approximately half of departments had a Twitter (X) account, but the median number of annual tweets per department was relatively low. The departments that had the most tweets tended to be larger departments with more NIH funding, implying access to more resources. Most tweets were related to promotion of departmental activities and individual faculty, while few tweets were related to anesthesiology as a specialty overall. There are likely opportunities for more academic anesthesiology departments to use Twitter (X) and social media more broadly in the future. Departments could also broaden their tweet content, perhaps describing their values, explaining how anesthesiologists contribute to patient care, and providing education about anesthesiology as a medical specialty to the public.

## Supporting information

**S1 File. Analytic dataset used for the study.**
(XLSX)

## Author Contributions

**Conceptualization:** Michael Mazzeffi, Lindsay Strickland, Zachary Coffman, Ebony Hilton, Lynn Kohan, Ryan Keneally, Peggy McNaull, Nabil Elkassabany.

**Data curation:** Michael Mazzeffi, Lindsay Strickland, Zachary Coffman, Braden Miller, Ebony Hilton.

**Formal analysis:** Michael Mazzeffi, Lindsay Strickland, Zachary Coffman, Braden Miller, Lynn Kohan, Peggy McNaull, Nabil Elkassabany.

**Investigation:** Michael Mazzeffi, Ebony Hilton.

**Methodology:** Michael Mazzeffi, Lindsay Strickland, Zachary Coffman, Braden Miller, Lynn Kohan, Ryan Keneally, Peggy McNaull, Nabil Elkassabany.

**Project administration:** Michael Mazzeffi, Lindsay Strickland, Zachary Coffman, Lynn Kohan, Nabil Elkassabany.

**Resources:** Michael Mazzeffi, Ryan Keneally, Peggy McNaull.

**Software:** Michael Mazzeffi.

**Supervision:** Michael Mazzeffi, Ebony Hilton, Lynn Kohan, Ryan Keneally, Peggy McNaull, Nabil Elkassabany.

**Writing – original draft:** Michael Mazzeffi, Lindsay Strickland, Zachary Coffman, Braden Miller, Ebony Hilton, Lynn Kohan, Ryan Keneally, Peggy McNaull, Nabil Elkassabany.

**Writing – review & editing:** Michael Mazzeffi, Lindsay Strickland, Zachary Coffman, Braden Miller, Ebony Hilton, Lynn Kohan, Ryan Keneally, Peggy McNaull, Nabil Elkassabany.

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
