## [Decision Letter · Decision Letter 0]

12 Dec 2023

PONE-D-23-29211Cross Sectional Study of Twitter (X) Use Among Academic Anesthesiology Departments in the United StatesPLOS ONE

Dear Dr. Mazzeffi,

Thank you for submitting your manuscript to PLOS ONE. After careful consideration, we feel that it has merit but does not fully meet PLOS ONE’s publication criteria as it currently stands. Therefore, we invite you to submit a revised version of the manuscript that addresses the points raised during the review process.

We look forward to receiving your revised manuscript.

Kind regards,

Robin Haunschild

Academic Editor

PLOS ONE

Journal Requirements:

3. In your Methods section, please include additional information about your dataset and ensure that you have included a statement specifying whether the collection and analysis method complied with the terms and conditions for the source of the data.

Reviewers' comments:

Reviewer's Responses to Questions

**Comments to the Author**

1. Is the manuscript technically sound, and do the data support the conclusions?

Reviewer #1: Partly

Reviewer #2: Yes

2. Has the statistical analysis been performed appropriately and rigorously? 

Reviewer #1: Yes

Reviewer #2: N/A

3. Have the authors made all data underlying the findings in their manuscript fully available?

Reviewer #1: Yes

Reviewer #2: Yes

4. Is the manuscript presented in an intelligible fashion and written in standard English?

Reviewer #1: Yes

Reviewer #2: Yes

5. Review Comments to the Author

Reviewer #1: Your paper has been written focusing on Anesthesiology which explains why your references are limited. I suggest you broaden the search and compare with other medical and dental specialties for better comparison.

Reviewer #2: The author researched the utilization of Twitter (X) in anethsology. That was interesting and fit to the current world perseption. Let me ask one question.

/Do you have any information about other platform of social media

/What did this research academically mean?

6. PLOS authors have the option to publish the peer review history of their article (what does this mean?). If published, this will include your full peer review and any attached files.

Reviewer #1: No

Reviewer #2: No

---

## [Author Response · Author response to Decision Letter 0]

28 Dec 2023

Journal Requirements:

Response: We have updated all style requirements based on the two links above.

Response: We have deposited the analytic dataset in this submission as a supporting information file.

3. In your Methods section, please include additional information about your dataset and ensure that you have included a statement specifying whether the collection and analysis method complied with the terms and conditions for the source of the data.

Response: We have added this into the manuscript.

Response: We have made the analytic dataset used for this study available as a supporting information file. There are no HIPAA related or privacy issues, as this study was non-human subjects’ research. The dataset is uploaded on OSF and a DOI is now provided.

Response: We are making the analytic dataset available as a supporting information file. The manuscript has been updated to reflect this. The dataset is uploaded on OSF and a DOI is provided now.

Response: This has been done.

Response: We have double checked all references for correctness.

Reviewers' comments:

Reviewer's Responses to Questions

Comments to the Author

1. Is the manuscript technically sound, and do the data support the conclusions?

Reviewer #1: Partly

Reviewer #2: Yes

Response: No change required.

2. Has the statistical analysis been performed appropriately and rigorously?

Reviewer #1: Yes

Reviewer #2: N/A

Response: No change required.

3. Have the authors made all data underlying the findings in their manuscript fully available?

Reviewer #1: Yes

Reviewer #2: Yes

Response: The analytic dataset has been made available as a supporting information file.

4. Is the manuscript presented in an intelligible fashion and written in standard English?

Reviewer #1: Yes

Reviewer #2: Yes

Response: No change required.

5. Review Comments to the Author

Reviewer #1: Your paper has been written focusing on Anesthesiology which explains why your references are limited. I suggest you broaden the search and compare with other medical and dental specialties for better comparison.

Response: We have extensively updated our discussion to be more comprehensive. We now talk about the use of social media, mostly Twitter (X) by surgeons, ER physicians, dentists, and nurses.

Reviewer #2: The author researched the utilization of Twitter (X) in anethsology. That was interesting and fit to the current world perseption. Let me ask one question.

/Do you have any information about other platform of social media

Response: No we only collected data from Twitter, which was itself a VERY large undertaking. We have added this as a limitation of the study in the Discussion section.

/What did this research academically mean?

Response: This is a fair question. Overall one of our major points is that social media has enormous potential to disseminate information, enhance collaboration, and provide education to both professionals and the public. At the present time most academic anesthesiology departments are primarily using Twitter (X) to promote their own departments and faculty and disseminate news. There are opportunities in our opinion to use the platform for other purposes including providing education to the public about what physician anesthesiologists do on a day-to-day basis. Also, Twitter (X) is a powerful tool to disseminate important research findings, which could be done more by academic anesthesiology departments. 

We have added content related to this into the Discussion.

6. PLOS authors have the option to publish the peer review history of their article (what does this mean?). If published, this will include your full peer review and any attached files.

Do you want your identity to be public for this peer review? For information about this choice, including consent withdrawal, please see our Privacy Policy.

Reviewer #1: No

Reviewer #2: No

Response: No change required.

---

## [Decision Letter · Decision Letter 1]

30 Jan 2024

Cross Sectional Study of Twitter (X) Use Among Academic Anesthesiology Departments in the United States

PONE-D-23-29211R1

Dear Dr. Mazzeffi,

We’re pleased to inform you that your manuscript has been judged scientifically suitable for publication and will be formally accepted for publication once it meets all outstanding technical requirements.

Kind regards,

Robin Haunschild

Academic Editor

PLOS ONE

Additional Editor Comments (optional):

Reviewers' comments:

Reviewer's Responses to Questions

**Comments to the Author**

1. If the authors have adequately addressed your comments raised in a previous round of review and you feel that this manuscript is now acceptable for publication, you may indicate that here to bypass the “Comments to the Author” section, enter your conflict of interest statement in the “Confidential to Editor” section, and submit your "Accept" recommendation.

Reviewer #1: (No Response)

2. Is the manuscript technically sound, and do the data support the conclusions?

Reviewer #1: Partly

3. Has the statistical analysis been performed appropriately and rigorously? 

Reviewer #1: Yes

4. Have the authors made all data underlying the findings in their manuscript fully available?

Reviewer #1: Yes

5. Is the manuscript presented in an intelligible fashion and written in standard English?

Reviewer #1: Yes

6. Review Comments to the Author

Reviewer #1: You did not update your reference list ... it has been 26 references with no change before or after the review.

7. PLOS authors have the option to publish the peer review history of their article (what does this mean?). If published, this will include your full peer review and any attached files.

Reviewer #1: No

---

## [Editor Report · Acceptance letter]

1 Feb 2024

PONE-D-23-29211R1 

PLOS ONE

Dear Dr. Mazzeffi, 

I'm pleased to inform you that your manuscript has been deemed suitable for publication in PLOS ONE. Congratulations! Your manuscript is now being handed over to our production team.

Kind regards, 

on behalf of

Dr. Robin Haunschild 

Academic Editor

PLOS ONE